# Action Model Learning from Noisy Traces: a Probabilistic Approach
## Paper ID #228

**Primary Keywords:** *(1) Learning*

## Abstract

We address the problem of learning planning domains from plan traces that are obtained by observing the environment states through noisy sensors. In such situations, approaches that assume correct traces are not applicable. We tackle the problem by designing a probabilistic graphical model where preconditions and effects of every planning domain operators, and traces' observations are modeled by random variables. Probabilistic inference conditioned by the observed traces allows our approach to derive a posterior probability of an atom being a precondition and/or an effect of an operator. Planning domains are obtained either by sampling or by applying the maximum a posteriori criterion. We compare our approach with a frequentist baseline and the currently available state-of-the-art approaches. We measure the performance of each method according to two criteria: reconstruction of the original planning domain and effectiveness in solving new planning problems of the same domain. Our experimental analysis shows that our approach learns action models that are more accurate w.r.t. state-of-the-art approaches, and strongly outperforms other approaches in generating models that are effective for solving new problems.

## Introduction

The problem of learning symbolic planning domains, aka *action model learning*, (Aineto, Celorrio, and Onaindia 2019; Grand, Pellier, and Fiorino 2020; Mordoch, Juba, and Stern 2023) consists of inducing a symbolic planning domain from a set of observations of the executions of domain actions. Most of the proposed approaches start from a set of plan traces (Arora et al. 2018), i.e., sequences of (partial) descriptions of the states of the environment interleaved by the executed actions. Moreover, state descriptions are (often implicitly assumed to be) the result of decoding the subsymbolic signals acquired through the agent's sensors.

Assuming that the interpretation of the signal leads to a semantically correct description of the environment's state is often unrealistic. This is particularly evident when sensors generate high-throughput data, such as images, audio signals, natural language, point clouds, etc. In these cases, the mapping from perceptions to symbolic states is typically provided by deep neural networks, which return the truth value of state variables starting from the signals. For example, starting from the picture of an object $book_1$, a neural network returns if the object is open or closed, i.e. $open(book_1) = true$ or $open(book_1) = false$.

In such situations, action model learning approaches that assume correct traces, such as (Yang, Wu, and Jang 2007; Cresswell and Gregory 2011; Cresswell, McCluskey, and West 2013; Stern and Juba 2017; Lamanna et al. 2021), are not applicable. There are existing approaches designed for handling noisy traces (Rodrigues, Gérard, and Rouveirol 2011; Mourão et al. 2012; Segura-Muros, Pérez, and Fernández-Olivares 2018; Grand, Pellier, and Fiorino 2020); however, we believe that the problem of action model learning from noisy plan traces cannot be considered fully addressed; especially when the noise level is particularly high, where current state-of-the-art approaches do not provide a satisfactory performance.

We posit that adopting a more principled approach involving a probabilistic model, which explicitly represents the components of the problem along with the associated hypotheses and priors, has the potential to provide superior results. Consequently, we design an approach, called Noisy Offline Learning of Action Models (NOLAM), that is based on a probabilistic graphical model, where preconditions and effects of every planning domain operators, and traces' observations are modeled by random variables. More precisely, for each operator, every atom involving a subset of object types that are involved by the operator is associated with a boolean random variable that models whether the atom belongs to the set of positive (or negative) preconditions of the operator; similarly for the sets of positive and negative effects of the operator. Therefore, NOLAM learns a probability distribution of the preconditions and effects of each operator, which is conditioned on the input noisy plan traces and maximum noise level. Probabilistic inference on the probabilistic model adopted by NOLAM, conditioned by the observed traces, allows NOLAM to derive a posterior probability of an atom being a precondition and/or an effect of a planning domain operator. Planning domains are obtained either by sampling or by applying the maximum a posteriori criterion.

The advantage of employing probabilistic models for solving the action model learning problem lies in their ability to explicitly represent the (in)dependency hypotheses between random variables (associated with operators preconditions and effects) and explicitly model prior knowledge about their distribution. For example, the probabilis-

tic model adopted by NOLAM considers the dependence between an atom being a precondition and/or an effect of an operator. Moreover, NOLAM also models the prior knowledge of an atom being true/false in a given state. This approach offers the flexibility to explore various adaptations of the probabilistic model of NOLAM by adjusting priors and (in)dependence hypotheses.

The practical effectiveness of the proposed approach has been proved in an extensive experimental analysis on 23 past International Planning Competitions (IPCs) domains (Vallati et al. 2015). We compare NOLAM with a frequentist baseline and currently available state-of-the-art approaches. We measure the performance of each approach according to two criteria: reconstruction of the original planning domain and effectiveness in solving new planning problems of the same domain. Our experimental analysis shows that NOLAM outperforms all other approaches in terms of both accuracy of the learned action models and in generating models that are effective for solving planning problems.

The paper is structured as follows, we firstly discuss existing approaches for learning action models offline from noisy plan traces. Next, we provide some basic notions of planning domains and formally define plan traces with noisy states. Then, we describe in detail the probabilistic model adopted by NOLAM in terms of random variables and (in)dependence assumptions. Afterward, we provide an ablation study of NOLAM where we compare different assumptions about the prior knowledge and different criteria for performing probabilistic inference conditioned by the observed traces. Finally, we perform a comparison between NOLAM, a baseline, and two state-of-the-art approaches, by evaluating the accuracy of the learned models and the performance in generating models that are effective for solving planning problems.

## Related Work

The action model learning problem has been widely studied under different assumptions on the environment states and agent actions (Zhuo et al. 2010; Aineto, Celorrio, and Onaindia 2019; Lamanna et al. 2021; Verma, Marpally, and Srivastava 2021; Juba and Stern 2022; Mordoch, Juba, and Stern 2023). We discuss related approaches by focusing on the ones that learn action models offline from an input set of noisy plan traces. In (Mourão et al. 2012), the authors propose a method, namely ALICE, for learning action models from traces with noisy and incomplete states. ALICE trains a set of classifiers for predicting the action effects, and then extracts an action model from the classifiers in the form of a set of rules. However, the method proposed in (Mourão et al. 2012) is tested only with large amounts of data (i.e. about 20000 transitions per domain) and small noise levels (i.e. from 0 to 0.05). On the contrary, NOLAM achieves high performance with fewer data (i.e. few hundred of transitions per domain) and works with a noise rate ranging from 0 to 0.4.

Another approach for learning action models from plan traces with noisy actions is proposed in (Zhuo and Kambhampati 2013). The proposed approach takes as input a set of plan traces with no intermediate states, computes a set of "candidate" action models consistent with the input traces, and defines the probability distribution of the "candidate"

action models conditioned on the input traces. They learn the parameters of such a probability distribution by means of a policy gradient algorithm (Sutton and Barto 2018). The approach in (Zhuo and Kambhampati 2013) is substantially different from NOLAM since it does not deal with partial traces where intermediate states are noisy. Learning action models from traces with noisy intermediate states is the aim of this work.

In (Segura-Muros, Pérez, and Fernández-Olivares 2018), authors propose planminer, a method for learning action models from traces with noisy and incomplete states. For each planning domain operator, they build a dataset consisting of the transitions where the operator appears and then learn a set of logical rules by applying an inductive learning algorithm on the operator dataset. The learned rules can be related to either preconditions or effects of the operator. Finally, the rules are converted to a PDDL (McDermott et al. 1998) action model.

The work by (Agravante, Kimura, and Tatsubori 2023) learns action models from traces (with possibly noisy states) by means of a neuro symbolic model that relies on inductive logic (Riegel et al. 2020). They model the preconditions and effects of an operator as a set of logical rules, where each atom is associated with a binary weight. Then, they learn the binary weights and translate the weighted rules into a PDDL action model. However, no experimental analysis shows how introducing noise in the intermediate states affects the performance of this method.

Another method for learning action models from traces with noisy states is proposed in (Rodrigues, Gérard, and Rouveirol 2011). The proposed method learns a set of logical rules associated with preconditions and effects of actions, and then incrementally specialize or generalize the learned rules as new transitions are given as input. However, the method in (Rodrigues, Gérard, and Rouveirol 2011) is evaluated on a single domain (i.e. blocksworld) for learning the preconditions and effects of a single operator from a few hundred operator transitions. Our approach does not require such a large amount of data, i.e. it learns operator preconditions and effects from a few dozen of operator transitions.

Most importantly, the majority of the above approaches is based on inductive logic, which makes these approaches strongly depend on the input plan traces and less robust to high levels of noise in the states of the traces. On the contrary, NOLAM does not have this limitation. Furthermore, NOLAM uses a novel approach based on a probabilistic graphical model that allows to express a variety of (in)dependence assumptions between preconditions and effects of the planning domain operators. All of the approaches above are evaluated in a limited number of IPC domains ranging from 1 to 6, using an error metric that does not differentiate between the completeness and correctness of the learned models. On the contrary, we empirically prove the effectiveness of NOLAM by conducting an extensive experimental evaluation on 23 IPC domains; we evaluate the learned action models by means of standard precision and recall metrics adopted in (Aineto, Celorrio, and Onaindia 2019; Lamanna et al. 2021), and the validity of the plans computed with the learned action models.

There is a set of approaches that learn symbolic action models from continuous high-dimensional observations such as RGB images (Kurutach et al. 2018; Konidaris, Kaelbling, and Lozano-Pérez 2018; Asai and Fukunaga 2018; Hafner et al. 2019; Asai 2019; Asai and Muise 2021; Sartor et al. 2023). These approaches map noisy perceptions into latent propositional states by means of deep neural networks, and learn action models where preconditions and effects are specified in terms of latent state variables. However, the learned models can be used only for planning in the latent space, and the latent states need to be decoded back to human comprehensible representations. NOLAM does not suffer from these drawbacks, and could in principle learn action models from continuous high-dimensional observations by integrating deep neural networks that map the continuous observations into (possibly noisy) symbolic states.

## Background

Let $\mathcal{P}$ be a set of predicates with associated arity, and $\mathcal{O}$ be a set of operator names with associated arity. Predicates and operators of arity $n$ are called $n$-ary predicates and $n$-ary operators. Given an $n$-tuple $\boldsymbol{x} = \langle x_1, \ldots, x_n \rangle$ of distinct symbols (constants or variables), let $\mathcal{P}(\boldsymbol{x})$ be the set of atomic formulas $p(x_{i_1}, \ldots, x_{i_m})$ obtained by applying the $m$-ary predicate $p \in \mathcal{P}$ to any $m$-tuple of symbols $\langle x_{i_1}, \ldots, x_{i_m} \rangle$ in $\boldsymbol{x}$ (with $1 \leq i_1, \ldots, i_m \leq n$). For instance, if $\mathcal{P}$ contains the single binary predicate on, and $\boldsymbol{x} = \langle x_1, x_2, x_3 \rangle$. Then, $\mathcal{P}(\boldsymbol{x}) = \{\text{on}(x_i, x_j) \mid 1 \leq i, j \leq 3\}$.

**Definition 1 (Action schema)** *An* action schema *for an $n$-ary operator name $op \in \mathcal{O}$ on the set of predicates $\mathcal{P}$ is a tuple $\langle \text{par}(op), \text{pre}^+(op), \text{pre}^-(op), \text{eff}^+(op), \text{eff}^-(op) \rangle$, where $\text{par}(op)$ is a tuple of variables, $\text{pre}^+(op)$, $\text{pre}^-(op)$, $\text{eff}^+(op)$, and $\text{eff}^-(op)$ are four sets of atoms in $\mathcal{P}(\text{par}(op))$.*

**Definition 2 (Ground action)** *The ground action $a = op(c_1, \ldots, c_n)$ of an $n$-ary operator name $op \in \mathcal{O}$ w.r.t. the constants $c_1, \ldots, c_n$ is the triple $\langle \text{pre}^+(a), \text{pre}^-(a), \text{eff}^+(a), \text{eff}^-(a) \rangle$, where $\text{pre}(a)$ (resp. $\text{pre}^-(a)$, $\text{eff}^+(a)$, $\text{eff}^-(a)$) is obtained by replacing the $i$-th parameter of $\text{par}(op)$ in $\text{pre}^+(op)$ (resp. $\text{pre}^-(op)$, $\text{eff}^+(op)$, $\text{eff}^-(op)$) with $c_i$.*

We use the term *lifted*, as the opposite of *grounded*, to refer to expressions and actions where constants have been replaced with parameters.

**Definition 3 (Action model)** *An action model $\mathcal{M}$ is a triple $\langle \mathcal{P}, \mathcal{O}, \mathcal{H} \rangle$ where $\mathcal{P}$ is a set of predicates, $\mathcal{O}$ is a set of operator names with their arity and, for every $op \in \mathcal{O}$, $\mathcal{H}$ is a function mapping an operator name $op$ into an action schema.*

**Definition 4 (Trace)** *A* trace *$t$ is a set of $n$ transitions $\{\langle s_i, a_i, s_i' \rangle\}_{i=1}^n$ where $s_i' = s_i \setminus \text{eff}^-(a_i) \cup \text{eff}^+(a_i)$.*

Given a plan $\pi = (a_1, \ldots, a_n)$, the *plan trace* of $\pi$ is a trace $t_\pi = \{\langle s_{i-1}, a_i, s_i \rangle\}_{i=1}^n$ where the transitions are generated by executing $\pi$ from $s_0$. Similarly, a set $\mathcal{T}$ of $m$ traces can be defined as $\bigcup_{i=1}^m t_i$.

**Definition 5 (Noisy trace)** *Given a trace $t$, a noisy trace $\hat{t}$ of $t$ is a set of $n$ transitions $\{\langle \hat{s}_i, a_i, \hat{s}_i' \rangle\}_{i=1}^n$ where $\hat{s}_i$ (resp. $\hat{s}_i'$) is obtained by changing the truth value of some atom in $s_i$ (resp. $s_i'$).*

Notice that the only difference between a trace $t$ and a noisy trace $\hat{t}$ generated from $t$ is in terms of truth values of state atoms, whereas $t$ and $\hat{t}$ shares the same set of ground actions. We denote by $\hat{\mathcal{T}}$ a set of noisy traces generated from $\mathcal{T}$, where every $\hat{t}_i \in \hat{\mathcal{T}}$ is a noise trace of $t_i \in \mathcal{T}$.

## Method

Our method is based on a probabilistic model that explains how a given set of observations, i.e. the state transitions in a set of noisy traces, can be generated by the execution of a set of operators with a certain action schema. Given a set of observations of the state transitions generated by executing an operator $op \in \mathcal{O}$, we want to find, for every atom $p(\boldsymbol{x}) \in \mathcal{P}(\text{par}(op))$, the probability that $p(\boldsymbol{x})$ is a positive/negative precondition and/or a positive/negative effect of $op$.

### Graphical model for noisy trace generation

We start by introducing the random variables that model the sets of preconditions and effects of an operator, the truth values of propositions in the state transitions generated by executing the operator, and the noisy truth values of propositions in the same state transitions observed through noisy sensors. Then, we describe the probabilistic model adopted by NOLAM for inferring the probability of an atom being a positive/negative precondition (or effect), given the truth values of such atom in the observed state transitions.

For each operator $op \in \mathcal{O}$ of an action model $\mathcal{M}$, and for every atom $p(\boldsymbol{x}) \in \mathcal{P}(\text{par}(op))$, we introduce two random variables $R_{op,p(\boldsymbol{x})}$ and $E_{op,p(\boldsymbol{x})}$ that take values in $\{+, -, \emptyset\}$. The variable $R_{op,p(\boldsymbol{x})}$ (resp. $E_{op,p(\boldsymbol{x})}$) indicates if $p(\boldsymbol{x})$ is a positive, a negative, or is not a precondition (resp. effect) of $op$. For example, $R_{op,p(\boldsymbol{x})} = +$ indicates that $p(\boldsymbol{x}) \in \text{pre}^+(op)$.

We make the following independence assumptions:

- For every pair of distinct operators $op, op' \in \mathcal{O}$ and for every atom $p(\boldsymbol{x})$, we assume that $R_{op,p(\boldsymbol{x})}$ and $R_{op',p(\boldsymbol{x})}$ are independent, similarly for $R_{op,p(\boldsymbol{x})}$ and $E_{op',p(\boldsymbol{x})}$, $E_{op,p(\boldsymbol{x})}$ and $E_{op',p(\boldsymbol{x})}$, and $E_{op,p(\boldsymbol{x})}$ and $R_{op',p(\boldsymbol{x})}$. This assumption is typically adopted in PDDL, where the specification of the sets of preconditions and effects of an operator does not depend on the sets of preconditions and effects of a different operator.

- For every operator $op \in \mathcal{O}$ and pair of distinct atoms $p(\boldsymbol{x}), p'(\boldsymbol{x}') \in \mathcal{P}(\text{par}(op))$, we assume that $R_{op,p(\boldsymbol{x})}$ and $R_{op,p'(\boldsymbol{x}')}$ are independent; $E_{op,p(\boldsymbol{x})}$ and $E_{op,p'(\boldsymbol{x}')}$ are independent, and $R_{op,p(\boldsymbol{x})}$ and $E_{op,p'(\boldsymbol{x}')}$ are independent. This assumption is rather strong; however, it is an acceptable trade-off between complexity and performance. Indeed, considering dependencies between multiple preconditions and effects would result in an exponential increase in the inference complexity. Notice that, we do not assume that $R_{op,p(\boldsymbol{x})}$ and $E_{op,p(\boldsymbol{x})}$ are independent.

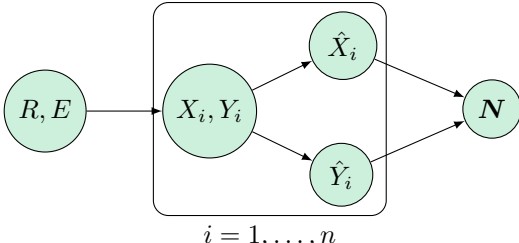

Figure 1: Graphical model showing the dependencies among the random variables adopted for modelling $Pr(\boldsymbol{N} \mid R, E)$.

The above assumptions allow us to describe NOLAM by focusing on a single operator $op \in \mathcal{O}$ and atom $p(\boldsymbol{x}) \in \mathcal{P}(\mathrm{par}(op))$, though NOLAM can be applied to a set of operators $\mathcal{O}$ and set of predicates applied to the operators parameters. For the sake of readability, in the following, we refer to $R_{op,p(\boldsymbol{x})}$ and $E_{op,p(\boldsymbol{x})}$ as $R$ and $E$, respectively.

The second group of random variables concerns the (observed) truth values of ground atoms in the state transitions obtained by executing an operator $op \in \mathcal{O}$. Let $\mathcal{T}$ be a set of $n$ transitions of $op$ and $\hat{\mathcal{T}}$ be a set of noisy traces generated from $\mathcal{T}$. For each $\langle s_i, op(\boldsymbol{c}_i), s'_i \rangle \in \mathcal{T}$, we introduce two boolean random variables $X_i$ and $Y_i$ that represent the truth values of the ground atom $p(\boldsymbol{c}_i)$ in $s_i$ and $s'_i$, respectively. For example, both $X_i$ and $Y_i$ being true indicates that $p(\boldsymbol{c}_i) \in s_i \cap s'_i$. Similarly, for the noisy transition $\langle \hat{s}_i, op(\boldsymbol{c}), \hat{s}'_i \rangle \in \hat{\mathcal{T}}$, we introduce two boolean random variables $\hat{X}_i$ and $\hat{Y}_i$ that represent the truth values of the ground atom $p(\boldsymbol{c}_i)$ in $\hat{s}_i$ and $\hat{s}'_i$, respectively. We model noise in the sensors by the conditional probabilities $P(\hat{X}_i \mid X_i)$ and $P(\hat{Y}_i \mid Y_i)$, which are two equal Bernoulli distributions with parameter $e \in [0, 1]$, i.e., $P(\hat{X}_i = \hat{x} \mid X_i = x)$ is equal to $e$ if $\hat{x} \neq x$ and $(1 - e)$ otherwise; similarly for $P(\hat{Y}_i \mid Y_i)$.

For each combination of values $(x, y) \in \{0, 1\}^2$, we introduce a random variable $N^{(x,y)}$ that counts the number of transitions in $\hat{\mathcal{T}}$ where $(\hat{X}_i, \hat{Y}_i) = (x, y)$:

$$N^{(x,y)} = \sum_{i=1}^{n} \mathbb{1}_{(\hat{X}_i, \hat{Y}_i) = (x,y)}. \qquad (1)$$

Let $\boldsymbol{N} = \langle N^{(1,1)}, N^{(1,0)}, N^{(0,1)}, N^{(0,0)} \rangle$ be a random vector of variables that take values in $\{0, \ldots, n\}$.

The dependencies among the previously introduced random variables are described in the graphical model in Figure 1. Notice that, since we do not assume $R$ and $E$ to be independent, we consider $(R, E)$ as a unique variable that takes values in $\{+, -, \emptyset\}^2$. Similarly, since $X_i$ and $Y_i$ are not independent, we consider $(X_i, Y_i)$ as a single variable that takes values in $\{0, 1\}^2$. The variables $X_i$ and $Y_i$ depend on $R$ and $E$ since, e.g., when $(R, E) = (+, -)$ then $(X_i, Y_i) = (1, 0)$; whereas $\hat{X}$ depends on $X$ since, with probability $1 - e$, $\hat{X} = x$ when $X = x$; similarly for $\hat{Y}$ and $Y$. Finally, $\boldsymbol{N}$ depends on $\hat{X}_i$ and $\hat{Y}_i$, as of Equation (1).

We assume that $P(X_i, Y_i \mid R, E)$ are equal for every $i$ in $\{1, \ldots, n\}$. Furthermore, $P(\hat{X}_i \mid X_i, Y_i) = P(\hat{X}_i \mid X_i)$

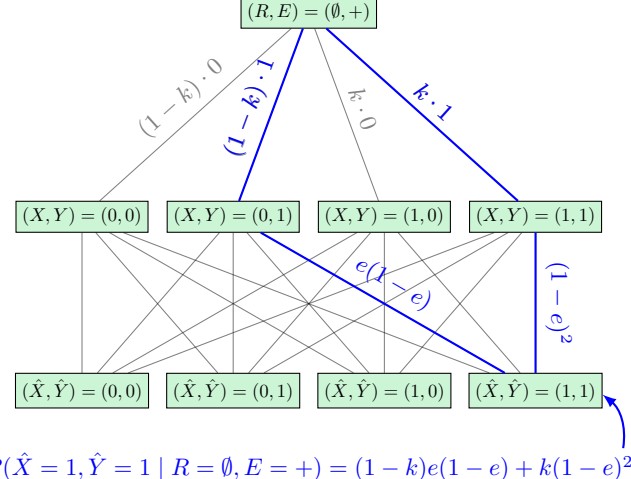

$$P(\hat{X} = 1, \hat{Y} = 1 \mid R = \emptyset, E = +) = (1 - k)e(1 - e) + k(1 - e)^2$$

Figure 2: Computing $P(\hat{X} = 1, \hat{Y} = 1 \mid R = \emptyset, E = +)$.

and $P(\hat{Y}_i \mid X_i, Y_i) = P(\hat{Y}_i \mid Y_i)$ since $\hat{X}_i$ (or $\hat{Y}_i$) only depends on $X_i$ (or $Y_i$). Finally, we assume that $P(\hat{X}_i \mid X_i) = P(\hat{Y}_i \mid Y_i)$, and they are equal for every $i$ in $\{1, \ldots, n\}$, i.e. the probability of correctly observing $\hat{X}_i$ (or $\hat{Y}_i$) does not change in different state transitions of $\hat{\mathcal{T}}$. This is reasonable when, e.g., the value of $\hat{X}$ is observed through a noisy sensor with a fixed noise level. Therefore, in the following we omit the index $i$ from $X_i, Y_i, \hat{X}_i$, and $\hat{Y}_i$. The independence of $P(X_i, Y_i \mid R, E)$ from $i$ implies that $Pr(\boldsymbol{N} = \boldsymbol{n} \mid R, E)$ factorizes as:

$$\binom{n}{\boldsymbol{n}} \prod_{x,y} Pr(\hat{X} = \hat{x}, \hat{Y} = \hat{y} \mid R, E)^{n^{(x,y)}} \qquad (2)$$

where $Pr(\hat{X}, \hat{Y} \mid R, E)$ is obtained by marginalizing $Pr(\hat{X} \mid X) Pr(\hat{Y} \mid Y) Pr(X, Y \mid R, E)$ over $X$ and $Y$, i.e.:

$$\sum_{x,y} Pr(\hat{X} \mid X = x) Pr(\hat{Y} \mid Y = y) Pr(X = x, Y = y \mid R, E)$$

We finally need to provide a model for $P(X, Y \mid R, E)$. Notice that $P(X, Y \mid R, E)$ is deterministic when $R$ is different from $\emptyset$. For instance, if $R = +$ and $E = -$, i.e. if $p(\boldsymbol{x})$ is a positive precondition and a negative effect, then $p(\boldsymbol{x})$ must be true before and false after executing the operator, and therefore $Pr(X = 1, Y = 0 \mid R = +, E = -) = 1$. When $R = \emptyset$ and $E = -$, i.e. $p(\boldsymbol{x})$ is neither a positive nor a negative precondition, $X$ can be either 0 or 1. The value of $Y$ instead is determined by $X$ and $E$. For instance, if $X = 1$ and $E = \emptyset$, then $Y$ must be 1. We therefore need to introduce a prior $k \in [0, 1]$ of $X$ being true. In our experiments, we set $k = 0.5$ since changing the value of $k$ had no significant impact on the performance.

**Example 1** *An example where we compute $P(\hat{X} = 1, \hat{Y} = 1 \mid R = \emptyset, E = +)$ is shown in Figure 2 and detailed in the following. We compute $Pr(\hat{X} = 1, \hat{Y} = 1 \mid R = \emptyset, E = +)$*

*by marginalizing over $X$ and $Y$:*

$$\sum_{x,y\in\{0,1\}} Pr(\hat{X}=1,\hat{Y}=1|X=x,Y=y) \cdot Pr(X=x,Y=y|R=\emptyset,E=+)$$

*Since $Pr(X=x,Y=0 \mid R=\emptyset,E=+)=0$ for every $x \in \{0,1\}$, we have that the above summation can be written as:*

$$Pr(\hat{X}=1,\hat{Y}=1 \mid X=0,Y=1)$$
$$\cdot Pr(X=0,Y=1 \mid R=\emptyset,E=+)$$
$$+ Pr(\hat{X}=1,\hat{Y}=1 \mid X=1,Y=1)$$
$$\cdot Pr(X=1,Y=1 \mid R=\emptyset,E=+)$$

*and the above sum is equal to:*

$$e(1-e)Pr(X=0,Y=1 \mid R=\emptyset,E=+)$$
$$+ (1-e)^2 Pr(X=1,Y=1 \mid R=\emptyset,E=+)$$

*which can be formulated as:*

$$e(1-e)Pr(X=0) + (1-e)^2 Pr(X=1)$$

*and we finally obtain:*

$$e(1-e)k + (1-e)^2(1-k)$$

*we can derive $Pr(\hat{X}=x,\hat{Y}=y \mid R=r,E=e)$ for $(x,y) \in \{0,1\}^2$ and $(r,e) \in \{+,-,\emptyset\}^2$ in a similar way. We report all possible values of $Pr(\hat{X},\hat{Y} \mid R,E)$ in the supplementary material.*

### Estimating preconditions and effects

Our objective is to estimate the *posterior* probability of $(R,E)$ given the observations $\boldsymbol{N}$, i.e. $Pr(R,E \mid \boldsymbol{N})$.

The *posterior* $Pr(R,E \mid \boldsymbol{N})$ can be exploited for obtaining an approximation of an action model that generates the traces $\mathcal{T}$. By the Bayes Theorem, $Pr(R,E \mid \boldsymbol{N})$ can be formulated as:

$$\frac{Pr(\boldsymbol{N} \mid R,E)Pr(R,E)}{Pr(\boldsymbol{N})} \qquad (3)$$

where $Pr(\boldsymbol{N} \mid R,E)$ is typically referred to as *likelihood*, whereas $Pr(R,E)$ is a *prior*, and $\mathcal{P}(\boldsymbol{N})$ is the probability of the observations $\boldsymbol{N}$.

The probability $Pr(\boldsymbol{N})$ can be computed by marginalizing over $R$ and $E$ as follows:

$$\sum_{r,e\in\{+,-,\emptyset\}} Pr(\boldsymbol{N},R=r,E=e). \qquad (4)$$

The joint probability $Pr(\boldsymbol{N},R,E)$ can be decomposed as:

$$Pr(\boldsymbol{N} \mid R,E)Pr(R,E) \qquad (5)$$

where the *prior* $Pr(R,E)$ is given as input, and the *likelihood* $Pr(\boldsymbol{N} \mid R,E)$ is computed according to Equation (2). After deriving $Pr(R,E \mid \boldsymbol{N})$, for each operator $op \in \mathcal{O}$ and $p(\boldsymbol{x}) \in \mathcal{P}(\mathsf{par}(op))$, NOLAM derives the sets $\mathsf{pre}_{op}^+, \mathsf{pre}_{op}^-, \mathsf{eff}_{op}^+$ and $\mathsf{eff}_{op}^-$ by applying the maximum a posteriori criterion on $Pr(R,E \mid \boldsymbol{N})$.

| Domain | #types | $|\mathcal{P}|$ | $|\mathcal{O}|$ | max arity $\mathcal{P}$ | max arity $\mathcal{O}$ |
|---|---|---|---|---|---|
| driverlog | 6 | 6 | 6 | 2 | 4 |
| n-puzzle | 2 | 3 | 1 | 2 | 3 |
| transport | 7 | 5 | 3 | 2 | 5 |
| tpp | 8 | 7 | 4 | 3 | 7 |
| hanoi | 3 | 3 | 1 | 2 | 3 |
| gripper | 4 | 4 | 3 | 3 | 4 |
| elevators | 6 | 8 | 6 | 2 | 5 |
| floortile | 4 | 10 | 7 | 2 | 4 |
| zenotravel | 6 | 4 | 5 | 2 | 6 |
| depots | 10 | 6 | 5 | 2 | 4 |
| ferry | 2 | 5 | 3 | 2 | 2 |
| satellite | 4 | 8 | 5 | 2 | 4 |
| spanner | 6 | 6 | 3 | 2 | 4 |
| gold-miner | 1 | 12 | 7 | 2 | 2 |
| nomystery | 6 | 6 | 3 | 3 | 6 |
| blocksworld | 1 | 5 | 4 | 2 | 2 |
| barman | 10 | 15 | 12 | 2 | 6 |
| parking | 2 | 5 | 4 | 2 | 3 |
| rover | 7 | 25 | 9 | 3 | 6 |
| matching-bw | 2 | 10 | 10 | 2 | 3 |
| sokoban | 3 | 4 | 2 | 3 | 5 |
| grid | 3 | 9 | 5 | 2 | 4 |
| miconic | 2 | 6 | 4 | 2 | 2 |

Table 1: Statistics of the IPC classical planning domains. For every domain (1st column), we report the number of object types (2nd column), the number of predicate names (3rd column), the number of operators (4th column), the maximum predicates arity (5th column), and the maximum operators arity (6th column).

## Experiments

We evaluate the accuracy of the action models learned by NOLAM from a set of noisy traces and the effectiveness of the learned action models for solving planning problems. We firstly perform an ablation study of NOLAM, and then a comparison with state-of-the-art approaches.

**Benchmarks** We conduct an experimental analysis on 23 classical planning domains taken from past IPCs. To generate the noisy traces used for learning, we proceeded as follows. For each domain, we generated 10 random problems using the generators available from past IPCs. The statistics about the domains are reported in Table 1. For every generated problem, we computed a solution plan with the *ground truth* action model provided in the IPCs. As a planner we adopted FastDownward (Helmert 2006) and a lazy greedy best-first search with context-enhanced additive (Eyerich, Mattmüller, and Röger 2012) and FastForward (Hoffmann 2001) heuristics. By executing every solution plan from the initial state of the solved problem, we obtained a set $\mathcal{T}$ of 10 traces for each domain. The generated traces contain a number of transitions ranging from 1 to 70, a number of objects ranging from 3 to 58, and a number of ground atoms ranging from 8 to 687. Finally, the set $\hat{\mathcal{T}}$ of noisy traces has been obtained by randomly changing the truth value

of each ground atom in every state in $\mathcal{T}$ with probability
$e \in \{0, 0.1, 0.2, 0.3, 0.4\}$.

**Evaluation metrics**   To evaluate the learned action models we use two main criteria. A first criterion measures the capability of *reconstructing* the ground truth model, by comparing the action model $\hat{\mathcal{M}}$, learned from the set of noisy traces $\hat{\mathcal{T}}$, with the ground truth model $\mathcal{M}$, which generated the set of traces $\mathcal{T}$. We compare the learned and ground truth action models using the well-known precision and recall metrics on the preconditions and effects of the operators. The precision $P_{\mathsf{pre}^+(op)}$ and recall $R_{\mathsf{pre}^+(op)}$ of the positive preconditions of an operator $op \in \mathcal{O}$ are defined as:

$$P_{\mathsf{pre}^+(op)} = \frac{|\hat{\mathsf{pre}}^+(op) \cap \mathsf{pre}^+(op)|}{|\hat{\mathsf{pre}}^+(op)|}$$

$$R_{\mathsf{pre}^+(op)} = \frac{|\hat{\mathsf{pre}}^+(op) \cap \mathsf{pre}^+(op)|}{|\mathsf{pre}^+(op)|}$$

where $\hat{\mathsf{pre}}^+(op)$ and $\mathsf{pre}^+(op)$ are the sets of positive preconditions of $op$ in $\hat{\mathcal{M}}$ and $\mathcal{M}$, respectively. Similarly, we define the precision and recall for the sets $\mathsf{pre}^-$, $\mathsf{eff}^+$ and $\mathsf{eff}^-$. The *overall* precision $P_{op}$ for an operator $op$ is defined by considering the sets $\mathsf{pre}(op) = \mathsf{pre}^+(op) \cup \mathsf{pre}^-(op)$ and $\mathsf{eff}(op) = \mathsf{eff}^+(op) \cup \mathsf{eff}^-(op)$:

$$P_{op} = \frac{|\hat{\mathsf{pre}}(op) \cap \mathsf{pre}(op)| + |\hat{\mathsf{eff}}(op) \cap \mathsf{eff}(op)|}{|\hat{\mathsf{pre}}(op)| + |\hat{\mathsf{eff}}(op)|}$$

$$R_{op} = \frac{|\hat{\mathsf{pre}}(op) \cap \mathsf{pre}(op)| + |\hat{\mathsf{eff}}(op) \cap \mathsf{eff}(op)|}{|\mathsf{pre}(op)| + |\mathsf{eff}(op)|}$$

We explicitly define $P_{op}$ instead of averaging $P_{\mathsf{pre}^+(op)}$, $P_{\mathsf{pre}^-(op)}$, $P_{\mathsf{eff}^+(op)}$, and $P_{\mathsf{eff}^-(op)}$, because for domain without negative preconditions $P_{\mathsf{pre}^-(op)}$ and $R_{\mathsf{pre}^-(op)}$ are not significant. However, we want to evaluate the impact of learning wrong negative preconditions. Therefore, we adopt the above definition of $P_{op}$ in our ablation study for measuring the impact of wrong predictions of negative preconditions. The *average* precision $P$ (or recall $R$) of $\hat{\mathcal{M}}$ w.r.t. $\mathcal{M}$ is defined by averaging $P_{op}$ (or $R_{op}$) for all the operators $op \in \mathcal{O}$.

A second criterion concerns the capability of the learned models of solving problems. There are situations where it is preferable to learn an action model capable of solving more problems, than an action model more similar to the ground truth model. Indeed, an error in the prediction of a precondition or an effect can have a substantially different impact on the learned model's capability of solving problems. For example, if the learned model has a precondition that is not in the ground truth model but is implied by another precondition in the ground truth model, such a precondition would not have a negative impact on solving problems effectively. Instead, a learned model that contains an effect that is not in the ground truth model may produce non-executable plans.

Therefore, we also evaluate the capability of a learned action model to generate *valid* plans, i.e. plans that are executable and actually achieve the goal according to the

| $V$ \ $v$ | $+$ | $-$ | $\emptyset$ |
|---|---|---|---|
| $R$ | $\frac{2(n^{(1,1)}+n^{(1,0)})}{3n}$ | $\frac{2(n^{(0,1)}+n^{(0,0)})}{3n}$ | $\frac{1}{3}$ |
| $E$ | $\frac{n^{(0,1)}}{n}$ | $\frac{n^{(1,0)}}{n}$ | $\frac{n^{(1,1)}+n^{(0,0)}}{n}$ |

Table 2: $Pr(V = v)$ with $V \in \{R, E\}$ and $v \in \{+, -, \emptyset\}$.

ground truth model. We measure such capability by computing two ratios EP and EV of solved problems and valid plans to a set of problems:

$$\text{EP} = \frac{\text{\# problems with a solution plan in } \hat{\mathcal{M}}}{\text{\# problems}}$$

$$\text{EV} = \frac{\text{\# problems with a solution plan in } \hat{\mathcal{M}} \text{ valid in } \mathcal{M}}{\text{\# problems}}$$

All the experiments have been conducted on a CPU Apple M1 Pro with 16 GB of RAM. [1]

**Ablation study**   We firstly perform an ablation study of NOLAM to investigate how different methods for deriving $\hat{\mathcal{M}}$ from $Pr(R, E \mid \boldsymbol{N})$ and different assumptions about the prior $Pr(R, E)$ affect the precision and recall of $\hat{\mathcal{M}}$.

We compare two methods for deriving $\hat{\mathcal{M}}$ from the posterior $Pr(R, E \mid \boldsymbol{N})$: maximum a posteriori (*MAP*) and sampling (*sample*).

We also consider an orthogonal analysis that distinguishes the situations where the absence of negative preconditions is given as input knowledge by setting $Pr(R = -, E) = 0$, and the situation where $Pr(R, E)$ admits also negative preconditions. We refer to the situation where $Pr(R = -, E) = 0$ by index $\mathsf{pre}^+$. The resulting variants of NOLAM are denoted by *MAP*, *sample*, *MAP*$_{\mathsf{pre}^+}$, and *sample*$_{\mathsf{pre}^+}$. We factorize the prior $Pr(R, E)$ as $Pr(R) \cdot Pr(E)$, where $Pr(R)$ and $Pr(E)$ are estimated from $\hat{\mathcal{T}}$ according to the formulas reported in Table 2. For example, the prior $Pr(E = +)$ is obtained as the ratio of transitions where the ground atom is false in the previous state and true in the destination state. It is worth noting that having $Pr(R = \emptyset) = 0$ would make every atom either a positive or a negative precondition. To avoid this situation, we set $Pr(R = \emptyset) = \frac{1}{3}$.

We report the average precision and recall of the action models learned by *MAP*, *MAP*$_{\mathsf{pre}^+}$, *sample* and *sample*$_{\mathsf{pre}^+}$ in Figures 3 and 4, respectively. The measures of precision and recall are averaged over 23 domains. Not surprisingly, *MAP*$_{\mathsf{pre}^+}$ (or *sample*$_{\mathsf{pre}^+}$) achieves higher precision than *MAP* (or *sample*), while the recall is comparable. This is because, for every domain, the ground truth model has no negative preconditions, and *MAP*$_{\mathsf{pre}^+}$ (or *sample*$_{\mathsf{pre}^+}$) cannot introduce negative preconditions in the learned model. However, we noticed that most of the negative preconditions learned by *MAP* and *sample* for a single operator are implied by some positive preconditions of such an operator. For example, in the blocksworld domain, the operator

---
[1]The code is available in the supplementary material

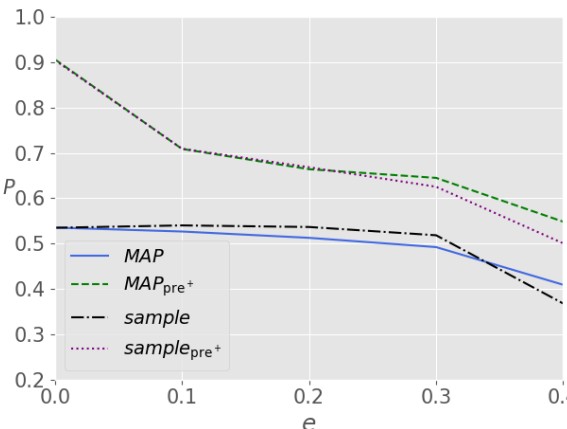
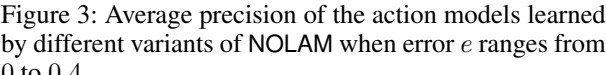

Figure 3: Average precision of the action models learned by different variants of NOLAM when error $e$ ranges from 0 to 0.4.

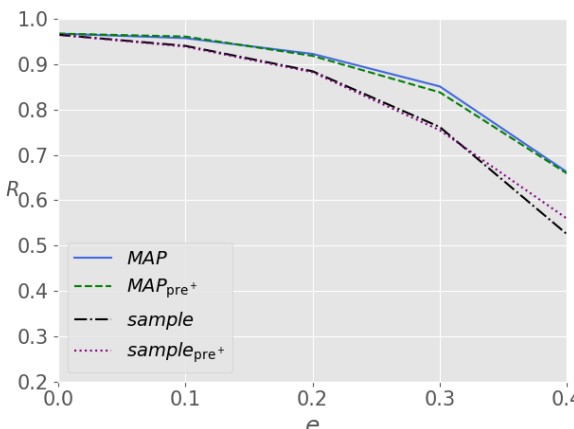

Figure 4: Average recall of the action models learned by different variants of NOLAM when error $e$ ranges from 0 to 0.4.

| | pre$^+$ | | | | eff$^+$ | | | | eff$^-$ | | | |
|---|---|---|---|---|---|---|---|---|---|---|---|---|
| | NOLAM | baseline | planminer | ALICE | NOLAM | baseline | planminer | ALICE | NOLAM | baseline | planminer | ALICE |
| $e$ | $P_{\text{pre}+}$ $R_{\text{pre}+}$ | $P_{\text{pre}+}$ $R_{\text{pre}+}$ | $P_{\text{pre}+}$ $R_{\text{pre}+}$ | $P_{\text{pre}+}$ $R_{\text{pre}+}$ | $P_{\text{eff}+}$ $R_{\text{eff}+}$ | $P_{\text{eff}+}$ $R_{\text{eff}+}$ | $P_{\text{eff}+}$ $R_{\text{eff}+}$ | $P_{\text{eff}+}$ $R_{\text{eff}+}$ | $P_{\text{eff}-}$ $R_{\text{eff}-}$ | $P_{\text{eff}-}$ $R_{\text{eff}-}$ | $P_{\text{eff}-}$ $R_{\text{eff}-}$ | $P_{\text{eff}-}$ $R_{\text{eff}-}$ |
| 0 | **0.83 0.97** | 0.79 **0.97** | 0.77 0.93 | 0.78 0.42 | **0.97 0.97** | **0.97 0.97** | 0.93 0.93 | 0.85 0.81 | **0.92 0.92** | **0.92** 0.91 | 0.84 0.87 | 0.80 0.74 |
| 0.1 | **0.82** 0.96 | 0.78 **0.97** | 0.66 0.54 | 0.70 0.39 | **0.96 0.97** | 0.95 0.95 | 0.52 0.70 | 0.69 0.72 | 0.90 **0.91** | **0.92** 0.90 | 0.40 0.57 | 0.66 0.61 |
| 0.2 | **0.79** 0.91 | 0.78 **0.96** | 0.56 0.59 | 0.56 0.40 | **0.92 0.95** | **0.92** 0.91 | 0.51 0.67 | 0.45 0.48 | 0.83 **0.87** | **0.84** 0.81 | 0.32 0.50 | 0.48 0.51 |
| 0.3 | **0.78** 0.82 | 0.75 **0.90** | 0.51 0.54 | 0.50 0.38 | **0.83 0.9** | 0.68 0.60 | 0.35 0.54 | 0.39 0.47 | **0.77 0.82** | 0.60 0.52 | 0.34 0.53 | 0.36 0.43 |
| 0.4 | **0.66** 0.70 | 0.61 **0.77** | 0.45 0.52 | 0.44 0.38 | **0.54 0.63** | 0.28 0.24 | 0.22 0.37 | 0.29 0.31 | **0.48 0.60** | 0.23 0.17 | 0.21 0.32 | 0.33 0.41 |

Table 3: Comparison of the average precision and recall for the positive preconditions and positive/negative effects of the action models learned from 10 traces with error $e$ ranging from 0 to 0.4. The precision and recall values are averaged over 23 domains. The average CPU time (in seconds) required for learning every single action model is $0.2s$ for NOLAM, $0.18s$ for baseline, $0.14s$ for ALICE and $7.23s$ for planminer.

PUT-DOWN$(x)$ has the positive precondition HOLDING$(x)$, indicating that to put down a block $x$ the agent must be holding $x$. However, when e.g. $e = 0$ then *MAP* and *sample* learn the negative precondition ¬HANDEMPTY$()$, indicating that, to put down $x$, the agent hand cannot be empty, which is implied by HOLDING$(x)$ being true. It is worth noting that learning negative preconditions implied by positive ones decreases the precision $P$ but does not affect the effectiveness of the learned action models for computing valid plans.

Interestingly, the recall of *MAP* (or $MAP_{\text{pre}+}$) is higher than the recall of *sample* (or $sample_{\text{pre}+}$), and the gap in terms of recall tends to increase as the error $e$ increases. These results show that the maximum a posteriori criterion provides action models that are more accurate in terms of recall and better or comparable in terms of precision than the sampling criterion, regardless of the admittance of negative preconditions.

**Comparison with state-of-the-art approaches** We compare NOLAM with a baseline and two state-of-the-art approaches[2]: planminer (Segura-Muros, Pérez, and Fernández-Olivares 2018) and ALICE (Mourão et al. 2012). Since plan-

miner and ALICE learn action models with negative preconditions, for a fair comparison, we adopt the *MAP* variant of NOLAM. We also compare with a frequentist baseline, which we refer to as baseline, that considers only the priors $Pr(R)$ and $Pr(E)$ defined in Table 2 and selects the values $v$ for $R$ and $E$ that maximize the priors.

We report the average precision and recall grouped by pre$^+$, eff$^+$ and eff$^-$ for error $e$ ranging from 0 to 0.4 (Table 3). The metrics $P_{\text{pre}-}$ and $R_{\text{pre}-}$ are not reported since the considered domains have no negative preconditions. NOLAM always outperforms planminer and ALICE in both precision and recall, in many cases, by a large margin (e.g. when $0.2 \leq e \leq 0.4$). On average, NOLAM improves the precision and recall w.r.t. planminer of 0.29 and 0.25, respectively; similarly, NOLAM improvements of precision and recall w.r.t. ALICE are respectively 0.25 and 0.36 on average. These results show that NOLAM learns action models that are much more accurate than the ones learned by current state-of-the-art approaches.

Surprisingly, our baseline achieves competitive performance w.r.t. NOLAM. The precision $P_{\text{pre}+}$ achieved by NOLAM is always greater than the one achieved by baseline. However, when $0.2 \leq e \leq 0.4$, the recall $R_{\text{pre}+}$ is better for baseline. We believe this is because baseline adds more

[2]For other approaches there is no available code, which prevents us from further comparisons

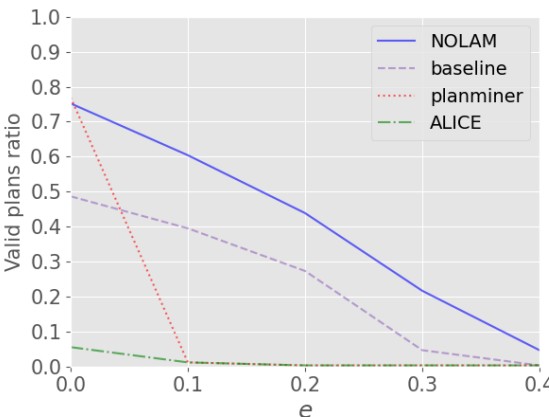

Figure 5: Average ratio of valid plans produced with the learned action models.

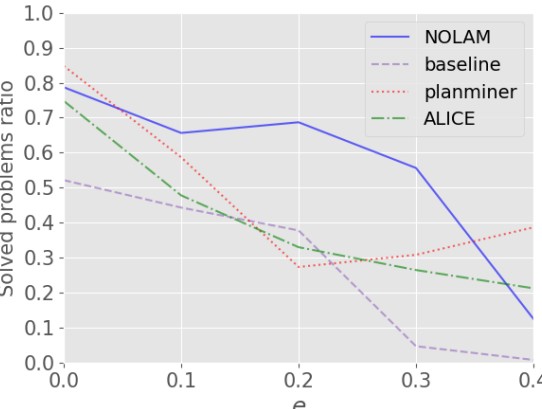

Figure 6: Average ratio of problems solved with the learned action models.

positive preconditions to every operator, introducing positive preconditions that are not in $\mathcal{M}$ (which reduces $P_{\text{pre}+}$), but reducing the number of positive preconditions that are in $\mathcal{M}$ and not in $\mathcal{M}$ (which increase $R_{\text{pre}+}$). The precision $P_{\text{eff}+}$ of NOLAM is better than or comparable to $P_{\text{eff}+}$ achieved by baseline for all considered values of $e$; similarly for $R_{\text{eff}+}$, $P_{\text{eff}-}$ and $R_{\text{eff}-}$. Notably, with high error $e$, i.e. $e \in \{0.3, 0.4\}$, NOLAM outperforms baseline in terms of (positive and negative) affects precision and recall by a large average margin (0.21 and 0.35, respectively). This is empirical evidence that, with high levels of noise, NOLAM is more robust than baseline in learning accurate action models.

We further evaluate the action models learned by NOLAM, baseline, planminer, and ALICE on their capability of solving new planning problems, i.e., problems that were not used for generating the set of traces $\hat{\mathcal{T}}$. For each domain, we randomly generated a new set of 10 problems, and solved them with the learned models. For solving each problem, we set a CPU time limit of 60 seconds, which was sufficient for solving every problem with the ground truth action model.

Figure 6 and 5 report the measures of EP and EV averaged over the 23 considered domains. We evaluated the validity of the plans by means of the validation tool adopted in past IPCs (Howey, Long, and Fox 2004).

NOLAM strongly outperforms ALICE in terms of EV, for all considered values of $e$; similarly, NOLAM strongly outperforms planminer for all considered values of $e$ but $e = 0$, where planminer learns action models that produce plans as valid as NOLAM ones. However, the valid plans ratio achieved by planminer drastically decreases when $e > 0$. Surprisingly, the EV achieved by ALICE is always lower than 0.06. We noticed that the action models learned by ALICE lack of static preconditions, even when $e = 0$, which is a possible reason that causes them to not produce valid plans. Interestingly, the failures of planminer and ALICE are not mainly due to the fact that they do not find solution plans. Indeed, the values of EP achieved by planminer and ALICE are always higher than 0.27 and 0.21, respectively.

The gap between EP and EV is lower for NOLAM and baseline than for ALICE and planminer. Hence, the action models learned by NOLAM and baseline produce a lower number of invalid plans than the action models learned by ALICE and planminer.

Notably, despite the performance of the action models learned by NOLAM and baseline are comparable in terms of precision and recall, the models learned by NOLAM are much more effective for solving planning problems. The improvement of NOLAM w.r.t. baseline over plans validity ranges from 0.04 to 0.26, and decreases as the error $e$ increases. Even though baseline achieves higher $R_{\text{pre}+}$ and higher/comparable $P_{\text{eff}-}$ and $P_{\text{eff}+}$ than NOLAM when $e \leq 0.2$, then EV is significantly higher for NOLAM than for baseline. This is an empirical evidence that metrics precision and recall can be misleading when evaluating the learned action models quality. These results show that the probabilistic model adopted by NOLAM allows to learn action models that are much more effective for solving planning problems than current state-of-the-art approaches.

## Conclusions and Future Work

We presented an approach for learning action models offline from an input set of plan traces with noisy states. Our approach, namely NOLAM, models the fact that an atom is a precondition or effect of an operator by means of a probabilistic model. NOLAM performs probabilistic inference given a set of observations consisting of the transitions in the plan traces. We compare NOLAM with two state-of-the-art approaches on 23 IPC planning domains, by evaluating the accuracy of the learned models and the effectiveness of the learned models for solving planning problems. In our experimental analysis, NOLAM strongly outperforms other approaches. In future work, we will extend NOLAM to an online setting, where the traces are generated online by executing action and observing the environment state through noisy sensors.

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
