# OpenReview forum: "Action Model Learning from Noisy Traces: a Probabilistic Approach"
_icaps-conference.org/ICAPS/2024/Conference — ICAPS 2024_

### Official Review · Reviewer_8oGX · 2024-01-18

**Significance And Importance:** 2
**Soundness:** 3
**Novelty:** 3
**Clarity:** 2
**Overall Evaluation:** 1
**Confidence:** 4

**Weaknesses:**

0: Minor weaknesses requiring some work to be addressed for the paper to be accepted.

**Contributions Of The Paper:**

This paper addresses the challenge of learning action models from noisy transitions. The authors propose a novel approach named NOLAM, which leverages a probabilistic graphical model to calculate posterior probabilities for the preconditions and effects of each action and variable, utilizing a set of noisy transitions as input. Subsequently, these probabilities are translated into an action model through either the maximum a posteriori estimate or sampling methods. The resulting action model undergoes evaluation in terms of accuracy, precision, and efficacy in solving planning problems, with comparisons made against alternative approaches. The empirical assessment shows NOLAM's superiority over other methods.

**Ethical Considerations:**

(1) Not Applicable: The paper does not have any ethical considerations to address

**Nomination For Best Paper:**

No

**Questions For Authors:**

1. Can you justify the assumptions of the paper, given the motivation?
2. Can you discuss the problem with respect to the PDDL requirements? (quantifies, conditional effects, etc.)
3. Do the trends reported in the results persist when the number of trajectories changes?
4. Can you conjecture on why the baseline solves less problems?

**Reproducibility:**

5: Code and domains (whichever apply) are already publicly available

**Strengths Of The Paper:**

The proposed approach is both straightforward to comprehend and implement. Furthermore, the results indicate that NOLAM excels in learning accurate action models, even when provided with only a limited number of trajectories, thereby surpassing the performance of existing methods.

**Weaknesses Of The Paper:**

While I eventually managed to understand the content of the paper, it required multiple readings of certain sections due to issues affecting clarity. Notably, numerous typos, some leading to confusion, were present. Additionally, a considerable number of sentences were excessively lengthy or lacked clarity. Specific examples of these problems are provided below. Furthermore, the problem at hand was not adequately defined. Particularly, the background section failed to formally specify what is an input to the learning problem and what is the output. I assume that the operator names and arity are given, and only the action schema is being learned, but it is never mentioned explicitly. Moreover, there was no explicit discussion regarding the scope of the problem with respect to PDDL capabilities. For instance, it is clear that numeric variables are not supported, but if I understand correctly, conditional effects are also not supported.

A second notable weakness pertains to the assumptions made in the paper. While the motivation convincingly establishes the necessity for learning from noisy transitions, particularly in scenarios where sensors are translated into symbolic representation using deep neural networks (DNNs), the paper makes a questionable assumption of a fixed noise level. This assumption lacks practicality in the context of DNNs, where the noise level is unlikely to remain constant, and is expected to drastically change in different areas of the state space. Moreover, the algorithm's evaluation falls short in examining more realistic scenarios where this assumption may not hold. Another assumption involves the independence of $\hat{X}_i$ and $\hat{Y}_i$. It is probable that the sensor would exhibit similar levels of noise in two consecutive states, especially a DNN-based sensor.

Finally, the empirical evaluation uses a fixed number of transitions (ten) and does not perform an ablation study to see the quality of the learning with a different number of input trajectories (maybe on the harder domains in the set). Personally, I don’t find the EP measure very interesting. “learning” an action model with no preconditions and all possible effects result in 100% EP. Finally, figures 3 and 4 should also show the std (e.g., as a shaded area around the mean) and not only the average. Finally, there is no explanation for why NOLAM solves more problems than the baseline, even when their P and R metrics are comparable. The authors mention that these metrics can be misleading, but do not provide a clear explanation for this phenomenon.

I am deliberating on whether to recommend accepting or rejecting the paper based on these weaknesses. At this point, I am leaning towards recommending acceptance; however, it's crucial to note that my decision is subject to change based on the author's response.

Detailed clarity issues:
1.	Line 41: “starting from the signals”. The term starting is confusing. Originally, I thought that you meant that a sequence of signals generates a state, but after additional reading, I now assume you mean “based on the signal”, or “using the signal as an input”.
2.	The sentence that spans from line 66 to line 72 (!) is very hard to follow.
3.	Line 238: triple = tuple
4.	Lines 276-281: long sentence that is hard to follow.
5.	Line 323: X_i and Y_i are not very indicative names. Why not use S_i and S’_i to indicate the meaning of the variables?
6.	Line 343: “similarly”. The word similarly is confusing, since R and E are dependent while X_i and Y_i are independent.
7.	Line 360 and Eq.2: you use n as the number of examples and \textbf{n} as the value of N. It is hard to differentiate between these symbols when they are in the same equation (e.g., n \choose \textbf{n}).
8.	Line 473: “Instead” should be “In contrast”.
9.	Line 520: “However, when e.g. e=0 then MAP” => “However, when e=0, for example, MAP and sample learn..”
10.	Figures 5 and 6: use the names EV and EP in the caption.
11.	Lines 565-566: “but reducing the number of positive preconditions that are in M and not in M”. I understand why the precision is decreased and the recall is increased based on the previous sentence, but I could not interpret what you meant here.
12.	Line 615: “that metrics precision and recall can” => “that the precision and recall metrics can”.

------------------------------------------------------------------------------------------------------------------------------------------------------
#Post rebuttal:
I appreciate the authors' response. However, certain weaknesses of the paper, especially regarding the assumption of a fixed noise level, were not fully addressed. I find this assumption unrealistic, particularly considering the motivation for utilizing DNNs to generate symbolic input from images. This concern is echoed by reviewer gLvw. Additionally, the authors' response regarding the effect of different numbers of trajectories on the results is not fully satisfactory. While they provided additional results, these only include their algorithm and lack a comparison with the baselines.

Despite these issues, I recognize the merit of this work. However, I suggest the authors adopt a more modest tone regarding their contribution, framing it as a step toward handling noisy trajectories rather than a comprehensive solution to the more realistic problem of dependent noise without prior information.

In summary, I maintain my weak acceptance recommendation but recommend that the authors refine their claims regarding the problem that their approach is capable of solving and provide a more in-depth analysis of the impact of varying the number of trajectories in the final submission.

---

> ### Author Rebuttal · Authors · 2024-01-26
>
> We thank the reviewer for the insightful feedback and valuable comments. In the following, we answer the Reviewer questions.
>
> Q1:
> [Fixed noise level]
> Compare the answer to Reviewer 1: In addition, dealing with a noise level that changes in different areas of the state space would require conditioning the noise model with a subset $Z$ of state variables, obtaining a noise model $Pr(\hat{X} | X,Z)$. However, $Z$ can be observed with noise, which would require a model of the form $Pr(\hat{X},\hat{Z}|X,Z)$. As for the noise changing in time (see answer to Reviewer 1). In principle, this can be done at the price of having the observations $\textbf{N}$ split into the different values of $Z$, and devising a way of aggregating the different predictions.
>
> [Independence of $\hat{X}$ and $\hat{Y}$]
> Notice that we assume  $\hat{X}$ and  $\hat{Y}$ to be *conditionally* independent, i.e. $Pr(\hat{X},\hat{Y}|X,Y) = Pr(\hat{X}|X,Y)*Pr(\hat{Y}|X,Y)$. In general, this does not imply that  $\hat{X}$ and  $\hat{Y}$ are independent.
> The fact that “the sensors exhibit similar levels of noise in two consecutive states” is implied by the assumption that noise model is the same in every state.
>
> Q2: We focus on classical planning domains, though it is simple to extend our approach to probabilistic effects, while extension to other PDDL requirements is out of the scope of this paper.
> As the Reviewer guesses, NOLAM requires as input the operator names, arity, and input object types. We will clarify it in the paper.
>
> Q3: To answer this question, we performed an additional experiment. We run the NOLAM variant used in Table 3 with a number of traces ranging from $1$ to $10$. For this analysis, we considered the hardest domain in Table 1, i.e., the domains with a number of predicates $\ge 10$. Due to the impossibility of attaching images, we report the precision $P$ and recall $R$ values of the learned models at the following anonymised link:
> https://drive.google.com/file/d/1NwCVqZu6_rXO0Nad_O6HWSyyipjrrPbe/view?usp=share_link
>
> As expected, the performance decreases as the number of traces decreases. Notably, the higher the error 'e' the more significant the decrease in performance when the number of traces decreases.
>
>
> Q4: We noticed that the baseline tends to add more positive preconditions than NOLAM, thus it learns action models with more false positive preconditions, which makes the learned models less effective for solving planning problems. We will clarify this in the paper.

---

### Official Review · Reviewer_gLvw · 2024-01-20

**Significance And Importance:** 2
**Soundness:** 3
**Novelty:** 2
**Clarity:** 3
**Overall Evaluation:** 1
**Confidence:** 5

**Weaknesses:**

0: Minor weaknesses requiring some work to be addressed for the paper to be accepted.

**Contributions Of The Paper:**

This paper designs a probabilistic model called NOLAM to address the problem of learning action models from an input set of noisy plan traces. NOLAM models preconditions and effects of operators, the truth values of propositions in the states of the transitions generated by executing the operators, and the noisy truth values of observations in the same state transitions as random variables. The planning models obtained with NOLAM are compared with those obtained with a frequentist baseline and current available SOTA approaches. Two performance metrics are applied: reconstruction of the original domain and effectiveness in solving new problems.

**Ethical Considerations:**

(1) Not Applicable: The paper does not have any ethical considerations to address

**Nomination For Best Paper:**

No

**Questions For Authors:**

I'd like if authors could clarify my observations and comments labeled in the review as (1) and (2).


**** POST-REBUTTAL ****

I thank the authors for their responses to my concerns. I would have expected, though, more concise answers to my specific questions. In particular, I think the paper should include a more straightforward explanation of how the MAP of R and E impacts the noise of the states and the limitations of assuming a fixed noise level. It is important that the paper clearly states the problem that NOLAM is capable of solving highlighting not only the advantages of the probabilistic model but also the limitations of the approach due to the noise/independence assumptions.

**Reproducibility:**

5: Code and domains (whichever apply) are already publicly available

**Strengths Of The Paper:**

The paper proposes a simple but effective probabilistic model to address the problem of learning action models from noisy traces. The paper is generally well-written and presents an exhaustive experimental evaluation. It adopts a more principled approach and it also provides superior results than other SOTA approaches addressing the same problem.

**Weaknesses Of The Paper:**

The second independent assumption is indeed rather strong (as acknowledged by the authors). It is unclear to me how the sets x and x' compare. Specifically, I wonder whether the two tuples of symbols are disjoint or if may they share some symbols. I am assuming that x and x' are tuples that can contain some common symbols to denote, for example, one precondition and one effect of an operator (e.g., in the blocksworld domain, op=unstack, p(x)=on(x1,x2), p'(x')=holding(x1)). How does this independence assumption affect the generation of a noisy trace \hat{t} from 't'? Does this mean that the sensor may observe either on(x1,x2) or (not (on(x1,x2)) in the pre-state of 't' and holding(x1) or (not(holding (x1)) in the post-state of 't' independently of each other? Additionally, authors claim they do not assume that that R_op,p(x) and E_op,p(x) are indepedent. I don't understand what it means to have the same atomic formula in the preconditions and effects of an operator unless it refers to a positive/negative formula in the preconditions and the reverse (negative/positive) formula in the effects (e.g. (not (holding ?param_1)) in the preconditions of operator pick-up, and (holding ?param_1) in the effects of pick-up). If this is the case, what does it mean they are not indepedent? That if (not (holding b1)) is observed in the pre-state of action pick-up(b1) then (holding b1) is observed in the post-state of pick-up(b1) and viceversa? I think this issue is very relevant to fully understand how the noisy traces are generated. (1)

The formalization is a little bit messy at some points:

1) A noise trace \hat{t} is defined over a set of 'n' transitions, the same number of transitions of the given trace 't'. Not sure if authors assume that a noise level is introduced in every transition or there may be some non-noisy transitions in a trace.
2) In the definition of the graphical model, the index 'i' is being used for two purposes: (a) to refer to the symbols of the m-tuple 'c_i' and (b) to the transitions of the set \hat{T}. Furthermore, sometimes the text says op(c_i) and others op(c).
3) If I understood well, the probability of correctly observing the truth values of the ground atoms of the pre-state and post-state of a transition of \hat{T} is the same. Consequently, the same probability is given to all the state transitions of one operator 'op'. Authors justify this by saying this is reasonable when the truth values of a state is observed through a sensor with a fixed noise level. But then this applies also to the pre-state and post-state of the transitions of every operator in the trace as I assume the same noise level is applied all along the trace. Moreover, since a state 's' in the author's approach is a full assignment of truth values to propositions, every si and s'i of all the transitions of a trace have the same size. My point is that if the level of noise is, for instance, e=0.3, then every proposition of X and Y in all the states of the plan trace will be a noisy proposition with probability 0.3 and non-noisy with probability 0.7 regardless the operator. (2)

In general, I think that the section devoted to the generation of noisy traces would benefit from clarification and, particularly, from a proper explanation of the implications of the adopted assumptions in the authors' approach.

The theoretical foundation of the prediction model is pretty straightforward. I miss some indication of the threshold of MAP that is used to derive the sets of positive/negative preconditions/effects.

I wish the authors had included elaboration on the application of the theoretical framework to their approach.

Some remarks concerning the experimental evaluation:

1) The sampling method to derive the learned model \hat{M} is never explained.
2) How is MAP used to derive the learned model? What MAP values are used to include or not a precondition/effect in an operator of \hat{M}?
3) A brief explanation of what a frequentist model is should be included.

Overall, the superior performance of NOLAM is shown in the results. I agree with the authors that the metrics of precision and recall can be misleading when evaluating the quality of the learned action models, and the reason is that they are syntactic metrics that measure some sort of distance to the ground-truth model. To complement a proper evaluation and introduce a semantic evaluation, authors include results of the number of solved problems by \hat{M} and how many of the solution plans are valid in the original model M. Still, this evaluation is dependent on the existence of the ground-truth model M. In many practical applications, we want to learn an action model from observations where no ground-truth model is available as it happens in many Machine Learning scenarios. In my opinion, evaluating the quality of a learned model wrt to a supposedly existing true model is a limitation of the approach.

---

> ### Author Rebuttal · Authors · 2024-01-26
>
> We thank the reviewer for the insightful feedback and valuable comments. In the following, we answer the Reviewer questions.
>
> Q1: The learned model $\hat{M}$ is derived by sampling according to $Pr(R,E|\textbf{N})$. In particular, for each operator $op$ and atom $p(x)$, we compute a probability distribution $Pr(R_{op,p(x)},E_{op,p(x)}=r,e|N)$ for every value $r,e$ in $(+,\emptyset,-)$, we sample $(R,E)$ from this distribution. Then, $p(x)$ (resp, $\lnot p(x)$) is added to the preconditions of $op$ if $R=+$ (resp, $R=-$). Similarly for the effects of $op$.
> For example, consider a specific operator $op$ and atom $p(x)$ where $Pr(R=+, E=-|\textbf{N}) = 0.7, Pr(R=\emptyset, E=-|\textbf{N}) = 0.3$, and $Pr(R,E|\textbf{N}) = 0$ otherwise; then we randomly sample $(R,E)$ in {$(+,-), (\emptyset, -)$} according to probabilities 0.7 and 0.3, respectively.  Notice that $Pr(R,E|\textbf{N})$ sums to $1$.
>
>
> Q2: The MAP values used for deriving preconditions and effects are the ones provided by $Pr(R,E|\textbf{N})$. By applying MAP to the previous example, we select  $(R,E)=(+,-)$, since it is associated with the highest probability. This will result in $p(x)$ being a positive precondition and a negative effect of $op$.

---

### Official Review · Reviewer_4QkZ · 2024-01-22

**Significance And Importance:** 2
**Soundness:** 4
**Novelty:** 3
**Clarity:** 4
**Overall Evaluation:** 2
**Confidence:** 4

**Weaknesses:**

1: Minor weaknesses that are easily fixable.

**Contributions Of The Paper:**

The paper addresses the problem of learning planning action models under the presence of observation noise. It proposes a graphical model where preconditions, effects, and observations are modelled by random variables, and uses probabilistic inference conditioned on observed traces to learn the models. The performance in terms of the precision and recall of learnt models, and the number of problems solved and validity of plans with the learnt models is evaluated over 23 IPC benchmark domains.

**Ethical Considerations:**

(1) Not Applicable: The paper does not have any ethical considerations to address

**Nomination For Best Paper:**

No

**Questions For Authors:**

1. Please explain what would be required in order to relax the assumptions that the noise model is identical for all variables and for all time steps.
2. Please correct or explain/define the n^(x,y) in equation (2).
3. Why is the precision for MAP/sample Pre+ only 0.9 with e=0. And are there particular domains that are responsible?
4. You say you could in principle integrate neural networks to NOLAM. Can you confirm that what you're talking about is taking a neural network that has been already trained to predict the value of atoms, and using these predictions as observations in  NOLAM, rather than having a neuro-symbolic model trainable end-to-end? And if you are talking about the latter, please breifly explain how this would work and how gradients would be propagated back?

**Reproducibility:**

5: Code and domains (whichever apply) are already publicly available

**Strengths Of The Paper:**

The approach is interesting, simple/elegant, new, well formalised.

The paper is very didactic and well-written. It carefully spells out the assumptions underlying the graphical model, and details the probability calculations step by step.

The paper has carefully thought of, convincing, and complete experiments.

The related work is complete and informative.

**Weaknesses Of The Paper:**

The main weakness that I see are that some of the assumptions are strong. In particular the assumption that the noise model is identical for all variables at all times does not seem very realistic. What would be involved in relaxing this assumption? I assume the fact that it is identical for all variables is easy to relax? Given it takes 0.2s to learn the model of an action, you could probably afford a bit more expressiveness and more computation time.


I only have minor comments:

* Definition 1, I's say par(op) is a n-tuple of variables (sorry for being pedantic).
* I cannot understand the n^{(x,y)} in equation (2).  n^{(x,y)} is not defined. Is it the value of the random variable N(x,y)? It's a bit confusing given that n is used in the equation for something else. Idem in table 2.
* typos: line 400: “and P(N) is the probability of …” (should be Pr(N)). Line 565: “that are in M and not in M” (I think you mean in \hat{M}and not in M).
* After equation 5: given that you have spelt things out in so much detail, it looks strange not to spell out the MAP criterion.
* Figure 3: I am surprised that the precision for MAP/sample Pre+ is only 0.9 with e=0. Why is that and are there particular domains that are responsible?
* If you have space in the final version, it would be interesting to show per domain accuracy results with extreme values of e.
* Supplementary material: why not use VAL as the validator?

---

> ### Author Rebuttal · Authors · 2024-01-26
>
> We thank the reviewer for the insightful feedback and valuable comments. In the following, we answer the Reviewer questions.
>
>
> Q1: The assumption that the noise model is identical for all predicates is not strictly required since we treat each predicate independently. We can use a specific noise model for each predicate $p(x)$. This results in using different models for $Pr(\hat X_{p(x)} | X_{p(x)} )$.
> Supporting different noise models at different time steps is not so straightforward. To this aim we have to add to the noise model a variable $T$ modelling the time step, obtaining $Pr(\hat X_{p(x)} | X_{p(x)}, T )$, and splitting the observations $\textbf{N}$ into the different values ${t \in T}$. Finding the “correct” way to aggregate the predictions at each time step into a single prediction requires further investigation. Still, we believe that assuming the noise model is uniform in time is reasonable.
>
>
> Q2: $n^{(x,y)}$ denotes the value of the random variable $N^{(x,y)}$. For example, $N^{(1,1)}=n^{(1,1)}$ where $n^{(1,1)}=2$ indicates we have observed two transitions in which the predicate was true in both the pre-state and the post-state. We will explicitly define $n^{(x,y)}$ in the paper.
>
>
> Q3: This is due to the presence of positive static predicates that are selected as positive preconditions since they are always true in the input plan traces.
>
>
> Q4: We confirm that the integration with neural networks refers to predicting the truth value of atoms using a pre-trained neural network. We will further clarify this point in the paper.

---

### Meta-Review · Area_Chair_RuMZ · 2024-02-06

**Recommendation:** Accept (Oral)
**Confidence:** 5

**Metareview:**

There is consensus on the merits of the paper to be accepted. Please, read the reviews and revise the paper accordingly.

**Ethical Considerations:**

(5) Excellent: The paper comprehensively addresses all of the applicable ethical considerations